

# An ensemble approach for imbalanced multiclass malware classification using 1D-CNN

Binayak Panda[1], Sudhanshu Shekhar Bisoyi[2] and Sidhanta Panigrahy[3]

[1] Department of Computer Science and Engineering, Institute of Technical Education and Research, Siksha 'O' Anusandhan (Deemed to be) University, Bhubaneswar, Odisha, India
[2] Department of Computer Science and Information Technology, Institute of Technical Education and Research, Siksha 'O' Anusandhan (Deemed to be) University, Bhubaneswar, Odisha, India
[3] Haas School of Business, University of California, Berkeley, Berkeley, CA, United States of America

Corresponding author
Sudhanshu Shekhar Bisoyi, sudhanshu.bisoyi@gmail.com

## ABSTRACT

Dependence on the internet and computer programs demonstrates the significance of computer programs in our day-to-day lives. Such demands motivate malware developers to create more malware, both in terms of quantity and variety. Researchers are constantly faced with hurdles while attempting to protect themselves from potential hazards and risks due to malware authors' usage of code obfuscation techniques. Metamorphic and polymorphic variations are easily able to elude the widely utilized signature-based detection procedures. Researchers are more interested in deep learning approaches than machine learning techniques to analyze the behavior of such a vast number of virus variants. Researchers have been drawn to the categorization of malware within itself in addition to the classification of malware against benign programs to examine the behavioral differences between them. In order to investigate the relationship between the application programming interface (API) calls throughout API sequences and classify them, this work uses the one-dimensional convolutional neural network (1D-CNN) model to solve a multiclass classification problem. On API sequences, feature vectors for distinctive APIs are created using the Word2Vec word embedding approach and the skip-gram model. The one-vs.-rest approach is used to train 1D-CNN models to categorize malware, and all of them are then combined with a suggested ModifiedSoftVoting algorithm to improve classification. On the open benchmark dataset Mal-API-2019, the suggested ensembled 1D-CNN architecture captures improved evaluation scores with an accuracy of 0.90, a weighted average F1-score of 0.90, and an AUC score of more than 0.96 for all classes of malware.

## INTRODUCTION

Information technology has a significant impact on our daily lives in the modern day. People of all ages use e-commerce, e-banking, e-healthcare, and other online services frequently to meet their everyday needs as the internet's accessibility has increased. Due to the sheer number of people who are exposed to the internet, malicious coders are motivated to explore all avenues for emotionally and financially exploiting victims.

Malware (a malicious application) is the main tool used in cyberattacks to carry out hostile actions on the computers of targeted or harmed users (*Aslan & Samet, 2020*).

The number and variety of malicious programs continue to grow, posing an ongoing challenge to antimalware vendors and researchers. According to statistics from the Kaspersky Security Bulletin Report 2021, there were 64,559,357 distinct malicious programs discovered between November 2020 and October 2021, consisting 20 different categories such as Backdoor, Trojan, etc (*Kaspersky, 2021*). According to SonicWall's cyber threat analysis, there are now 5.5 billion known instances of malware, growing 2% annually by 2022 (*SonicWall, 2023*). It would be accurate to claim that the number of malware programs is increasing tremendously, but not the variety. Again, the resulting harms to people and businesses are getting worse every day. Therefore, research aimed at classifying malware can help in malware detection and mitigation.

The most popular method of malware detection is signature-based detection, which involves searching for a certain signature in a previously built signature store in order to label a program as malicious (*Shijo & Salim, 2015*). The majority of anti-malware vendors employ this approach, which stores signatures of previously identified malware for malware detection, albeit the signature store may be updated often. However, it is possible for both newly obfuscated malware and previously identified malware to go undetected (*Ucci, Aniello & Baldoni, 2019*). The other way for identifying malware by looking at the execution time parameters and related behaviors is behavior-based detection. The identification of both old and new unknown malware is better with behavior-based method than with signature-based method, albeit there may be a time-space trade-off between the two (*Gibert, Mateu & Planes, 2020*). In these situations, the researchers are compelled to look for rational, practical, and cutting-edge methods in order to identify unidentified malware. A behavioral study using several machine learning algorithms has demonstrated the potential for improving malware identification and classification (*Ucci, Aniello & Baldoni, 2019*; *Tayyab et al., 2022*). Compared to machine learning algorithms deep learning models are becoming more and more popular, but the time and resource requirements for model training continue to be a problem (*Liu et al., 2017*; *Alom et al., 2019*). Deep learning also performs well in the domain of information security, in addition to applications for image analysis and language processing (*Tekerek, 2021*).

To investigate the categorisation of each class of malware in this study, the application program interface (API) sequences from various malware classes are used as the characteristics. This work focuses on methods for improving multiclass classification results when working with unbalanced dataset. To combine the results of various one-dimensional convolutional neural network (1D-CNN) classifiers trained using the one-vs-rest idea, a 1D-CNN based ensembled architecture is provided. Training and testing are done with the help of the data collection Mal-API-2019 from *Catak et al. (2020)*.

The remainder of the article's contents are organized as follows: works by other researchers that are connected to the study are discussed in Related Work. The preprocessing of the dataset, followed by the algorithm, and other specifics of the suggested design are explained in the Ensembled 1D-CNN Architecture section. Comparison of the results and

performance-related graphs are shown in the Experimental Setup and Results section. The conclusions are highlighted in the final section.

## RELATED WORK

The discipline of malware analysis has benefited from the work of numerous researchers. For this area of study, there is an enormous amount of literature. The suggested method uses convolutional neural network (CNN) to classify malware into multiple classes. The relevant literature is examined in relation to the classification of malware into several families using CNN and other techniques.

*Vinod et al. (2010)* performed dynamic analysis on four types of metamorphic malware and produced malware signatures tracing their API sequences. In their experimental setting, they employed the API sequences of a total of 80 viruses from four families and 20 benign programs. They have calculated the degree of membership of a malware to a malware class using the chi-square test. They achieved accuracy of 80%, 80%, 75%, and 75% for the respective families G2, MPCGEN, IL SMG, and NGVCK using this method. Additionally, they suggested that greater precision may be attained by increasing the number of samples. The method of signature matching is vulnerable to newly discovered malware samples.

Control flow graphs and API call graphs were extracted by *Mehra, Jain & Uppal (2015)* using 600 instances of malware and 150 benign samples. The desired features have been extracted from API call graphs using their suggested Gourmand Feature Selection technique. They performed classification using the WEKA tool and achieved accuracy of 89%, 92.24%, 94.56%, 99.10%, and 91.08%, respectively, using the KNN, VP, NB, J-48, and SMO classifiers. They didn't parse API sequences sequentially; instead, they exclusively used portable executables.

*Zhang et al. (2016)* suggested a simple malware classification system using ensemble learning, using data from the Microsoft malware classification challenge of kaggle. It successfully assigned malware samples from the unbalanced training dataset to the appropriate family. *Kolosnjaji et al. (2016)* considered system call sequences to classify malware. They have extracted best features using convolutional and recurrent network layers. They have achieved average precison of 85.6% and recall of 89.4% using this hybrid neural network architecture.

*Han et al. (2019)* used the TF-IDF technique to examine the relationship between API calls in API sequence on 807 benign and 3,027 malicious (both packed and unpacked variation) samples. In order to identify and categorize malware, they applied the machine learning techniques Random Forest, Decision Tree, KNN, and XGBoost on their developed explainable malware detection framework (MalDAE). They used static, dynamic, and fused API sequences. They were able to reach accuracy of 84.96%, 79.65%, 74.74%, and 83.15% using dynamic API sequence, which is better than static API sequence. However, with fused API sequence, the accuracy increased to 94.39%, 88.42%, 85.26%, and 93.33%, respectively. Due to the difficulty in dealing with the large number of malware variations, they claimed that deep learning approaches could increase productivity. Using system call sequences from malicious and benign Android applications, *Xiao et al. (2019)* showed high recall of 96.6% and low FPR of 9.3%.

A dataset of 7107 API sequences for eight different classes of malware was produced and published by *Catak et al. (2020)*. In order to do multiclass classification, they have additionally trained single layer long short-term memory (LSTM), two layers LSTM, DT, KNN, RF, and SVM on the dataset. In comparison to all other models, they achieved Recall and Precision of 0.47 using single layer LSTM. *Li & Zheng (2021)* used this dataset to classify malware utilizing API calls using LSTM and gated recurrent unit (GRU) models. They have achieved recall of 0.58 and 0.59 for LSTM and GRU, respectively, with precision of 0.56 for both approaches. *Demirkiran et al. (2022)* classified malware families using the same benchmark dataset as *Catak et al. (2020)*. They have contrasted the Transformer, CANINE-S, and BERT models with their proposed model RTF. With an F1-score of 0.61 and an AUC score of 0.88, the RTF model outperformed all other models.

*Vasan et al. (2020)* performed image-based malware classification using an ensemble of pretrained CNN models, VGG16 and ResNet-50, and the open dataset Malimg. Comparing the ensembling approach to conventional ML-based models, it demonstrated excellent accuracy. To solve the uneven size of the malware files of the employed datasets BIG2015 and DumpWare10, *Tekerek (2021)* created a method called CycleGAN and the B2IMG method to convert binaries to images. Their experimental findings demonstrate greater accuracy compared to other CNN-based algorithms. *Catak et al. (2021)* created images from binaries using their dataset. The final dataset is applied to CNN for classification after dataset enhancement with additive noise approaches and picture augmentation. According to experimental findings, a dataset that includes noise factor has a classification accuracy of 0.96 for malware classes, which is higher than the accuracy of 0.83 for a dataset that does not include noise factor. By using recurrent neural network (RNN) and CNN, *Sun & Qian (2021)* have performed static analysis of the visual malware images used in their RMVC approach. Even with a small training dataset, they discovered accuracy greater than 92%. They mentioned testing their method's efficacy in dynamic analysis as a potential future direction. Using malware images made from malware binaries, *Hammad et al. (2022)* performed malware classification. The Malimg dataset is used to train and evaluate k-nearest neighbors (KNN), support vector machine (SVM), and extreme learning classifier (ELM) models. Features are retrieved using GoogleNet (a deep learning model) and Tamuar (a texture feature that correlates to human visual perception). They discovered that ELM performed better than any other model. They have recommended that data augmentation be used, which could enhance classification outcomes.

All prior authors have worked on LSTM applying text vectorisation approaches such as TF-IDF, BERT, and CANINE with the open dataset Mal-API-2019. Again, from the literature, it is found that CNN-based models perform better for sequential data classification. *Kavak et al. (2021)* also emphasized the use of current social theories to build new theoretical constructs for designing behavioral models that can deal with cybersecurity challenges.

This study examines the semantic connections between APIs in API sequences using the Word2Vec embedding method and the Skip-gram model. The 1D-CNN classifiers trained on the dataset using the one-vs-rest classification method are combined in the proposed

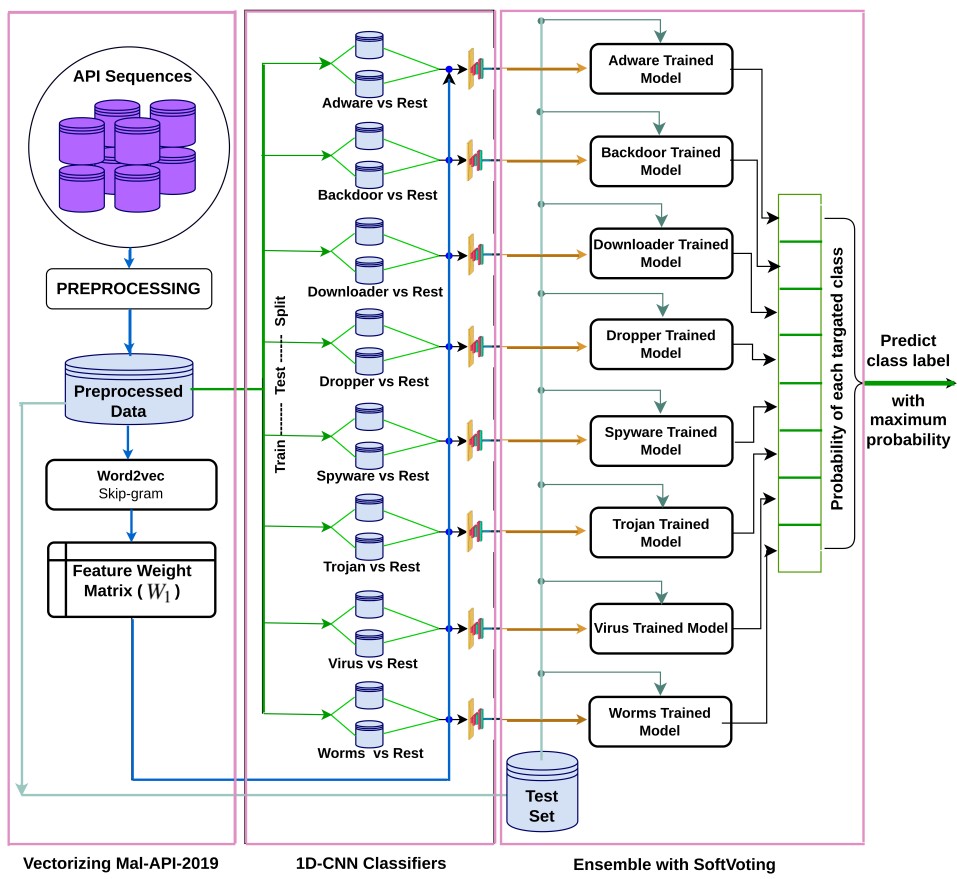

**Figure 1**    **Proposed ensembled 1D-CNN architecture.**

ensembled 1D-CNN architecture. The output from each classifier is integrated with the suggested ModifiedSoftVoting method, which looks at efficiency improvement.

# METHODOLOGY

## Proposed ensembled 1D-CNN architecture

In this study, the suggested architecture presented in Fig. 1 ensembles eight separately trained 1D-CNN models to address the multiclass malware classification problem. The mentioned architecture comprises three phases and utilises the *Mal-API-2019* dataset for training and testing. In the first phase, the dataset is vectorised using the Word2Vec Skip-gram model, and weight vectors are assigned to various APIs after investigating the semantic relationships between the APIs in the API sequences. In the second phase, eight 1D-CNN models are trained as One-vs-rest classifiers to learn one class against all others. The final phase uses the fundamental concept of combining classification abilities to classify individual types against all others. To combine all of those models and solve this multiclass classification problem, a *ModifiedSoftVoting* algorithm is suggested.

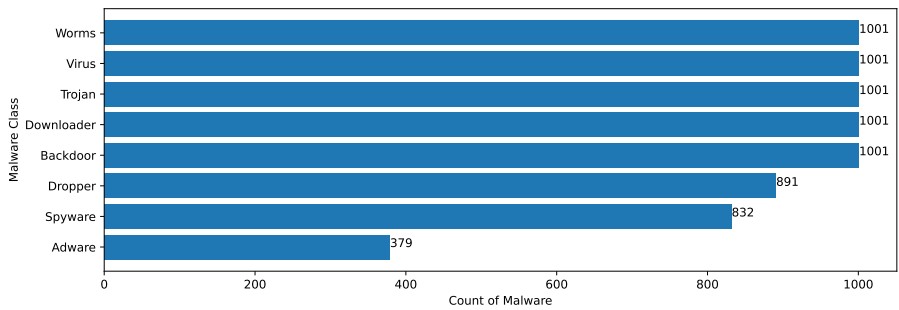

**Figure 2** Malware distribution class wise.

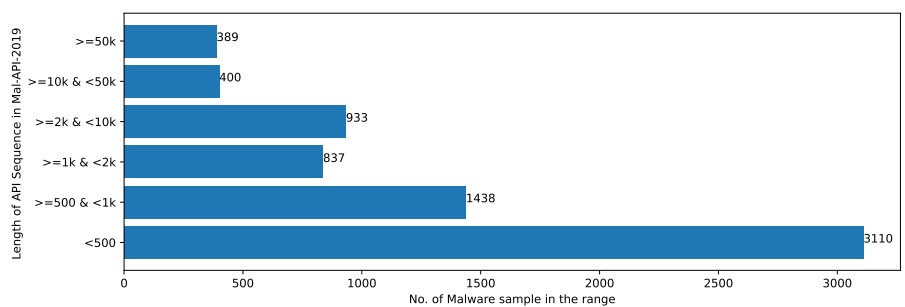

**Figure 3** Sample distribution before preprocessing.

### Phase-I: vectorizing Mal-API-2019

*Dataset Description:* The Mal-API-2019 dataset consists of API sequence records of 7107 malwares of eight different classes. Figure 2 speaks about the frequency distribution of each class and Fig. 3 shows the frequency of malware samples considering the API sequence length before preprocessing.

Figures 2 and 3 shows that Mal-API-2019 is highly imbalanced. Such diverse distribution has adverse effect on the success of classification. The said dataset is interpreted as having two atrributes named $API_{seq}$ and $M_{class}$. The former represents the sequence of APIs called by the malware and the latter represents the class of that malware. Each entry in $API_{seq}$ is interpreted as a sentence of type $M_{class}$ made up of finite number of words *i.e.,* APIs with repetitions from the $API_{Vocabulary} = \{API_0, API_1, API_2, ...., API_n\}$. The $M_{class}$ ranges from 0 to 7 representing eight different type of senetences maping to 8 malware classes. For the representation of textual documents in a multidimensional vector space, the vector space model is particularly popular. In one of the previous work TF-IDF vectorization technique has been used on DLL sequences for host based anomaly detection (*Panda & Tripathy, 2020*). In a comparable manner, word embedding vector of each distinct API of $API_{Vocabulary}$ is created using Skip-gram model of Word2Vec embedding technique. The redundant API occurrences are eliminated in order to address the dataset's variability with regard to the length of each entry. The distribution of records in the processed dataset against record length following duplicate removal is shown in Fig. 4.

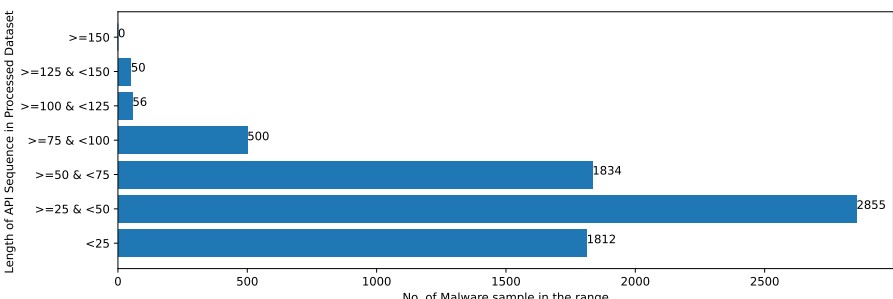

**Figure 4   Sample distribution after preprocessing.**

*Finding Word Embeddings for each API:* The word embeddings are discovered using Skip-gram by taking into account all the records of $API_{seq}$ for each unique API. Skip-gram does well with small datasets and depicts less common words better than more frequent ones (*Mikolov et al., 2013*). With the aim of capturing the embeddings (feature vectors) of each unique API that qualify the meaning of the API they represent, a skip-gram neural network model, as shown in Fig. 5, is utilized. Such feature vectors can be quite useful in describing the $M_{class}$ type. For each target API, the skip-gram technique takes into account all windows of size $\ell$ in order to extract the semantics of the APIs into embeddings. The embedding matrix $W_1$ shown in Fig. 5, which is given to the 1D-CNN models for training purposes in Phase-II of the proposed architecture, serves as the word embeddings for all different APIs. The weight matrix $W_2$ can be employed to predict the likelihood of various words given a context word. Since the main goal of this effort is to obtain the word embeddings, $W_2$ is not utilized.

### Phase-II: training 1D-CNN models

Due to their effectiveness, deep learning models are becoming more and more popular. The convolution layer, pooling layer, and fully connected artificial neural network layer (Dense Layer) are the three basic layers that make up the one-dimensional convolutional neural network (1D-CNN), a deep learning model. These models use convolution and pooling operations to learn features from sequential input, such as texts, and then conduct binary or multiclass classification in the dense layer. It performs the convolution operation with various kernels on the spatial input texts in the convolution layer to produce corresponding one-dimensional feature maps.

$$x_j^l = f\left(\sum_{i=1}^{M} x_i^{l-1} * k_{ij}^l + b_j^l\right) \tag{1}$$

The operation of the one-dimensional convolution layer is as described in Eq. (1), where $k$ and $j$ represents convolution kernels and number of kernels respectively. $M$ denotes channel number in input $x^{l-1}$ with $b$ as the bias to the corresponding kernel. The $*$ is the convolution operator and $f()$ is the activation function. The pooling layer uses the avarage pooling or max pooling method with a predetermined window to reduce the feature dimension produced by the convolution operation. Output of the final pooling

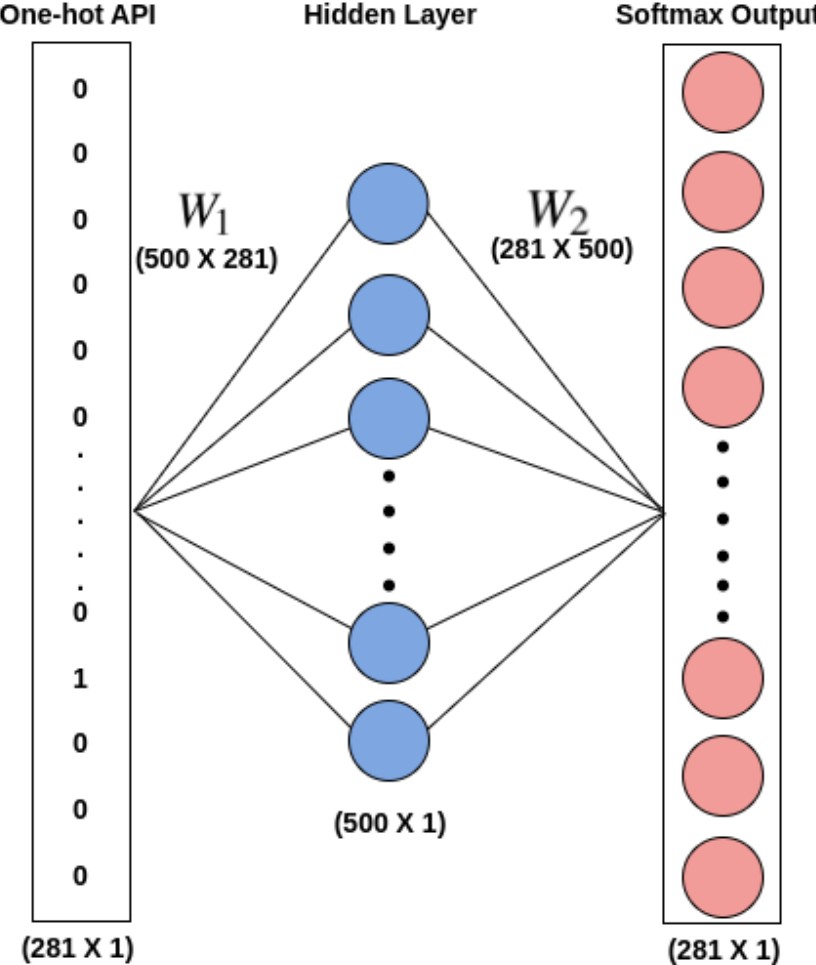

**Figure 5** Word2Vec skip gram model.

layer $l+1$ is given as input to the dense layer and the output of the dense layer is evaluted as described in Eq. (2), where $w$ and $b$ denotes weight and bias respectively.

$$h(x) = f\left(w^{l+1}.x^{l+1} + b^{l+1}\right) \tag{2}$$

The architecture and description of the 1D-CNN model utilised in this study for training and testing are shown in Fig. 6. Following the procedures outlined in Algorithm-1, each 1D-CNN classifier identified as $model_c$ in Fig. 1 is trained using $Mal\_API_c$ an intermediate dataset of the processed dataset. The statements 2-13 in Algorithm-1 generates the set *OvR_Mal_API_Datasets*, which is a collection of intermediate datasets. Every intermediate dataset present in *OvR_Mal_API_Datasets* labels the record containing the "class of interest" as 1 (positive) and the "rest all" as 0 (negative). The statements 14-20 trains 1D-CNN classifiers corresponding to each of the intermediate dataset present in *OvR_Mal_API_Datasets* and returns *1D_CNN_Classifier_List*, a list of classifiers. The third phase of the architecture combines the capabilities of all of these trained models.

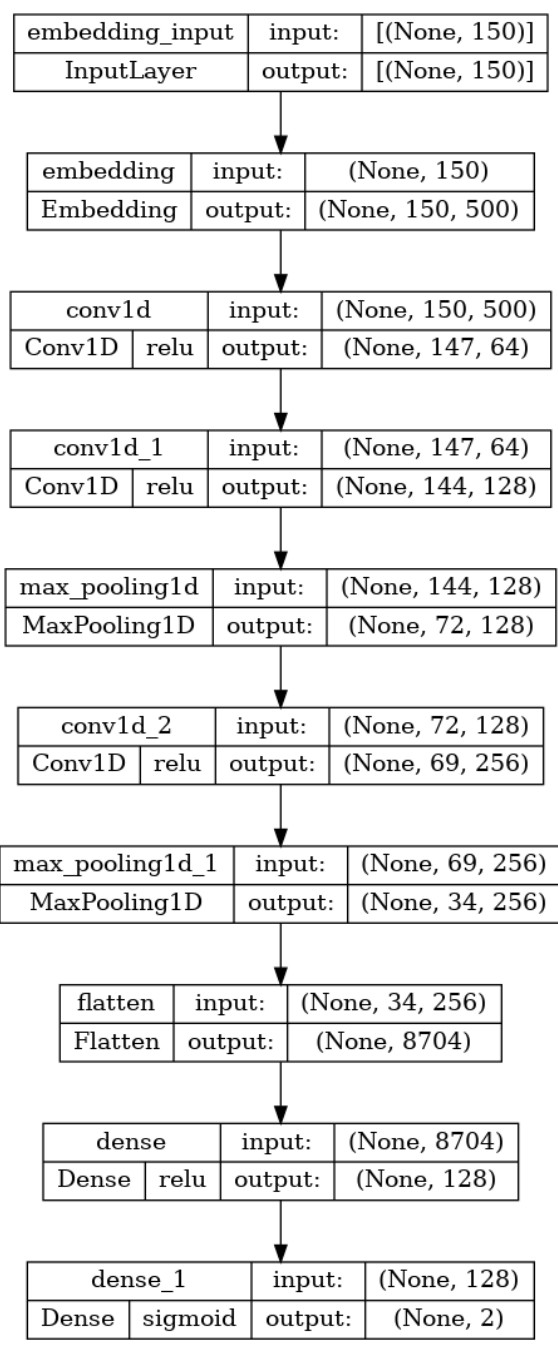

**Figure 6  Summary of the 1D-CNN model.**

### Phase-III: ensemble with SoftVoting

Using the *ModifiedSoftVoting* method outlined in Algorithm-2, the classification abilities of independently trained classifiers in the 1D_CNN_Classifier_List are ensembled. In this phase a *Test Set* is created from the processed dataset using stratified sampling as shown in statement 2 of the algorithm. Based on the outcomes of all the trained classifiers

for each record in the *Test Set*, this algorithm determines the best predicted class. In the $1D\_CNN\_Classifier\_List$, each classifier ($model_c$) predicts two probability scores of each record in the *Xtest* as '*not in class of interest*' and '*in class of interest*' respectively as per statements 3-7 of the algorithm. *ClassificationScore*( $c_{score}$) is a two dimensional array with a size of ($mX2$) where $m$ represents count of records in *Xtest* . The $i^{th}$ row of $c_{score}$ represents probability score for the $i^{th}$ record of *Xtest* as '*not in class of interest*' and '*in class of interest*' at *index 0 and 1* respectively. *ClassProbability*( $c_{proba}$) is a two dimensional array of size ($mX8$) where $m$ represents count of records in *Xtest*. The $j^{th}$ column of $c_{proba}$ will contain $m$ probability scores of $m$ records of *Xtest* as '*in class of interest*' using $j^{th}$ classifier $model_j$ in $1D\_CNN\_Classifier\_List$. The $i^{th}$ row of $c_{proba}$ represents probability scores of $i^{th}$ record of *Xtest* as '*in class of interest*' for all *classifiers* in $1D\_CNN\_Classifier\_List$ respectively.

---

**Algorithm 1** 1D-CNN Classifiers

---

**Require:** $W_1$, ProcessedDataset{ $API_{seq}$, $M_{class}$ }
**Ensure:** $1D\_CNN\_Classifiers\_List$

1: $EmbeddingMatrix = W_1$
2: $OvR\_MalAPI\_Datasets = \phi$
3: **for each** distinct class $c \in M_{class}$ **do**
4:     $Mal\_API_c = \phi$
5:     **for each** Record $\{API_{seq}, M_{class}\} \in ProcessedDataset\{API_{seq}, M_{class}\}$ **do**
6:         **if** ($M_{class} == c$) **then**
7:             $Mal\_API_c = Mal\_API_c \cup \{API_{seq}, 1\}$
8:         **else**
9:             $Mal\_API_c = Mal\_API_c \cup \{API_{seq}, 0\}$
10:         **end if**
11:     **end for**
12:     $OvR\_MalAPI\_Datasets = OvR\_MalAPI\_Datasets \cup \{Mal\_API_c\}$
13: **end for**
14: $1D\_CNN\_Classifiers\_List = \phi$
15: **for each** $Mal\_API_c \in OvR\_MalAPI\_Datasets$ **do**
16:     $Xtrain_c, Xtest_c, Ytrain_c, Ytest_c = train\_test\_split(Mal\_API_c[API_{seq}],$ $Mal\_API_c[M_{class}], 0.8)$
17:     $model_c = Conv1D\_Model(Xtrain_c, Ytrain_c, EmbeddingMatrix, validation\_data$ $= (Xtest_c, Ytest_c))$
18:     $1D\_CNN\_Classifiers\_List = 1D\_CNN\_Classifiers\_List \cup \{model_c\}$
19: **end for**
20: Return $1D\_CNN\_Classifiers\_List$

---

---

**Algorithm 2** ModifiedSoftVoting

---

**Require:** $1D\_CNN\_Classifier\_List$ as $C_{list}$, ProcessedDataset{ $API_{seq}$, $M_{class}$ }

**Ensure:** confusion_matrix as $cm$

1: Set $cm[n][n] = 0$            ▷ n: Number of distinct class in $M_{class}$

2: $Xtest, Ytest = test\_split(ProcessedDataset\{API_{seq}, M_{class}\})$     ▷ Test Set of multiclass
     records

3: **for each** $c \in C_{list}$ **do**

4:      $c_{index} = C_{list}.index(c)$            ▷ $c_{index}$: Index of classifier $c$ in $C_{list}$

5:      $c_{score} = c.predict(Xtest)$           ▷ $c_{score}$: Predicted ClassificationScore

6:      $c_{proba}[:c_{index}] = c_{score}[:1]$    ▷ Column 1 of $c_{score}$ is assigned to column $c_{index}$ of $c_{proba}$

7: **end for**

8: **for each** $r \in c_{proba}$ **do**           ▷ Row wise traversal on $c_{proba}$

9:      $r_{index} = c_{proba}.index(r)$         ▷ $r_{index}$ is the row index of r in $c_{proba}$

10:      $c_{plabel} = indexof(max(r))$     ▷ $c_{plabel}$:Predicted label as index of largest element in $r$

11:      $c_{tlabel} = Ytest[r_{index}]$          ▷ $c_{tlabel}$: Actual label at $r_{index}$ in $Ytest$

12:      $cm[c_{tlabel}, c_{plabel}] = cm[c_{tlabel}, c_{plabel}] + 1$

13: **end for**

14: Return $cm$

---

Statements 8-13 in Algorithm-2 constructs the confusion matrix($cm$) by ensembling all the predictions. The $cm$ is a square matrix of dimention $(nXn)$ where $n$ is the count of distinct classes in $M_{class}$. It contributes to the estimation of several performance metrics as mentioned in Eqs. (3), (4), (5) and (6). The fundamental parameters required to calculate various performance metrics are *TP*, *FN, FP,* and *TN*. These parameters for a specific class of interest is interpreted in the $cm$ as a case of:

a) **TP (True Positive):** When malware of "class of interest" is predicted as "class of interest"

b) **FN (False Negative):** When malware of "class of interest" is predicted as some other class.

c) **FP (False Positive):** When some other class of malware is predicted as malware of "class of interest"

d) **TN (True Negative):** When malware of other class is predicted as malware of other class.

Tables 1 and 2 represents two exemplary cases of confusion matrix and TP, FP, FN, and TN parameters for specific "class of intertest" $C_3$ and $C_2$ respectively in the context of multiclass problem.

***Accuracy*** of the model is estimated as count of correct predictions divided by total count of predictions. Equation (3) mathematically represents the calculation of accuracy. Sometimes accuracy may mislead, hence the performance is ensured by calculating average of precision, recall and $F_1$ score respectively for all classes in multiclass classification

**Table 1  When class of interest is Class-3.**

|  | $C_1$ | $C_2$ | $C_3$ | $C_4$ |
|---|---|---|---|---|
| $C_1$ | TN | FP | | TN |
| $C_2$ | TN | FP | | TN |
| $C_3$ | FN | FN | TP | FN |
| $C_4$ | TN | | FP | TN |

Actual / Predicted

**Table 2  When class of interest is Class-2.**

|  | $C_1$ | $C_2$ | $C_3$ | $C_4$ |
|---|---|---|---|---|
| $C_1$ | TN | FP | TN | |
| $C_2$ | FN | TP | FN | FN |
| $C_3$ | TN | FP | TN | |
| $C_4$ | TN | FP | TN | |

Actual / Predicted

problem.

$$Accuracy = \frac{\sum_{i=1}^{n} cm[i,i]}{\sum_{i=1}^{n} \sum_{j=1}^{n} cm[i,j]} \tag{3}$$

$$Precision_c = \frac{cm[c,c]}{\sum_{i=1}^{n} cm[i,c]} \tag{4}$$

$$Recall_c = \frac{cm[c,c]}{\sum_{i=1}^{n} cm[c,i]} \tag{5}$$

$$F_{1c} = \frac{2}{\frac{1}{Recall_c} + \frac{1}{Precision_c}} \tag{6}$$

Precision for a specific class c (**Precision$_c$**) is estimated to see the impact of *FP* as higher concern than *FN*, as explained in Eq. (4). It is estimated as the number of true positives devided by the number of predicted positives. Recall for a specific class c (**Recall$_c$**) is estimated to see the impact of FN as higher concern than FP. It is estimated as the number of true positives divided by total number of actual positives. $F_1$-Score of a specific class c (**F$_{1c}$**) is the harmonic mean of **Precision$_c$** and **Recall$_c$**. It is used to ensure high precision against high recall. Weighted and macro average of precision, recall and $F_1$-score are used as perfomace metrics for multiclass problems. Unweighted mean of each of these performance metrics are referred as macro average measure. Weighted mean of each of these performance metrics are referred as weighted average measure using count of samples of each class as the weight.

**Table 3** Classification report of 1D-CNN models.

| OvR_Mal_API_Datasets | 1D_CNN_Classifiers_List | | Precision | Recall | f1-score | Accuracy |
|---|---|---|---|---|---|---|
| $MAL\_API_0$ | $model_0$ | Rest(0) | 0.99 | 0.99 | 0.99 | 0.98 |
| | | Adware(1) | 0.88 | 0.83 | 0.85 | |
| $MAL\_API_1$ | $model_1$ | Rest(0) | 0.91 | 0.95 | 0.93 | 0.88 |
| | | Backdoor(1) | 0.61 | 0.46 | 0.52 | |
| $MAL\_API_2$ | $model_2$ | Rest(0) | 0.95 | 0.97 | 0.96 | 0.93 |
| | | Downloader(1) | 0.78 | 0.66 | 0.72 | |
| $MAL\_API_3$ | $model_3$ | Rest(0) | 0.95 | 0.95 | 0.95 | 0.91 |
| | | Dropper(1) | 0.65 | 0.63 | 0.64 | |
| $MAL\_API_4$ | $model_4$ | Rest(0) | 0.93 | 0.95 | 0.94 | 0.89 |
| | | Spyware(1) | 0.52 | 0.43 | 0.47 | |
| $MAL\_API_5$ | $model_5$ | Rest(0) | 0.90 | 0.93 | 0.91 | 0.85 |
| | | Trojan(1) | 0.46 | 0.36 | 0.40 | |
| $MAL\_API_6$ | $model_6$ | Rest(0) | 0.95 | 0.96 | 0.95 | 0.92 |
| | | Virus(1) | 0.73 | 0.70 | 0.72 | |
| $MAL\_API_7$ | $model_7$ | Rest(0) | 0.93 | 0.94 | 0.93 | 0.89 |
| | | Worm(1) | 0.60 | 0.56 | 0.58 | |

## EXPERIMENTAL SETUP AND RESULTS

The experimental work for the described "Ensembled 1D-CNN architecture" is carried out using "*Intel(R) Xeon(R) CPU E5-2620 v4 @ 2.10 GHz* " HPC with 128 GB of RAM, "*NVIDIA Corporation GP102 [GeForce GTX 1080 Ti]* " GPU, *Ubuntu-18.04 LTS*, and *Python 3.8*.

In Phase-I of the architecture, the word embedding matrix $W_1$ *[500 X 281]* is constucted for a total of 280 disinct APIs using Skip-gram model. The final word embeddings of each distinct API in embedding matrix $W_1$ is found using window size $\ell = 10$, vector size $= 500$ after considering several combinations of window size and vector size.

To address the imbalanced multiclass malware classification problem, a number of 1D-CNN models are trained and validated with the one *vs.* rest classification principle in Phase-II of the architecture. Figure 6 depicts the best configuration of the 1D-CNN model, which is decided by working around several way of consideration of convolution layers, MaxPolling layers, and dense layers with various parameters such as filters, kernel size, pool/window size, batch size, activation functions. *Adam* is found as the best optimizer after working around multiple optimizers like *Adam, Adaboost and Adadelta*. The training and validation of each $model_c$ respective to $M_{class}$ is done with *80:20* stratified split ratio of their respective dataset $MAL\_API_c$ from *OvR_MalAPI_Datasets*. *OvR_MalAPI_Datasets* is a set of eight datasets corresponding to eight models constructed using Algorithm-1. The performance metrics of each $model_c$ respective to malware classes Adware, Backdoor, Downloader, Dropper, Spyware, Trojan, Virus, and Worms are mentioned in Table 3.

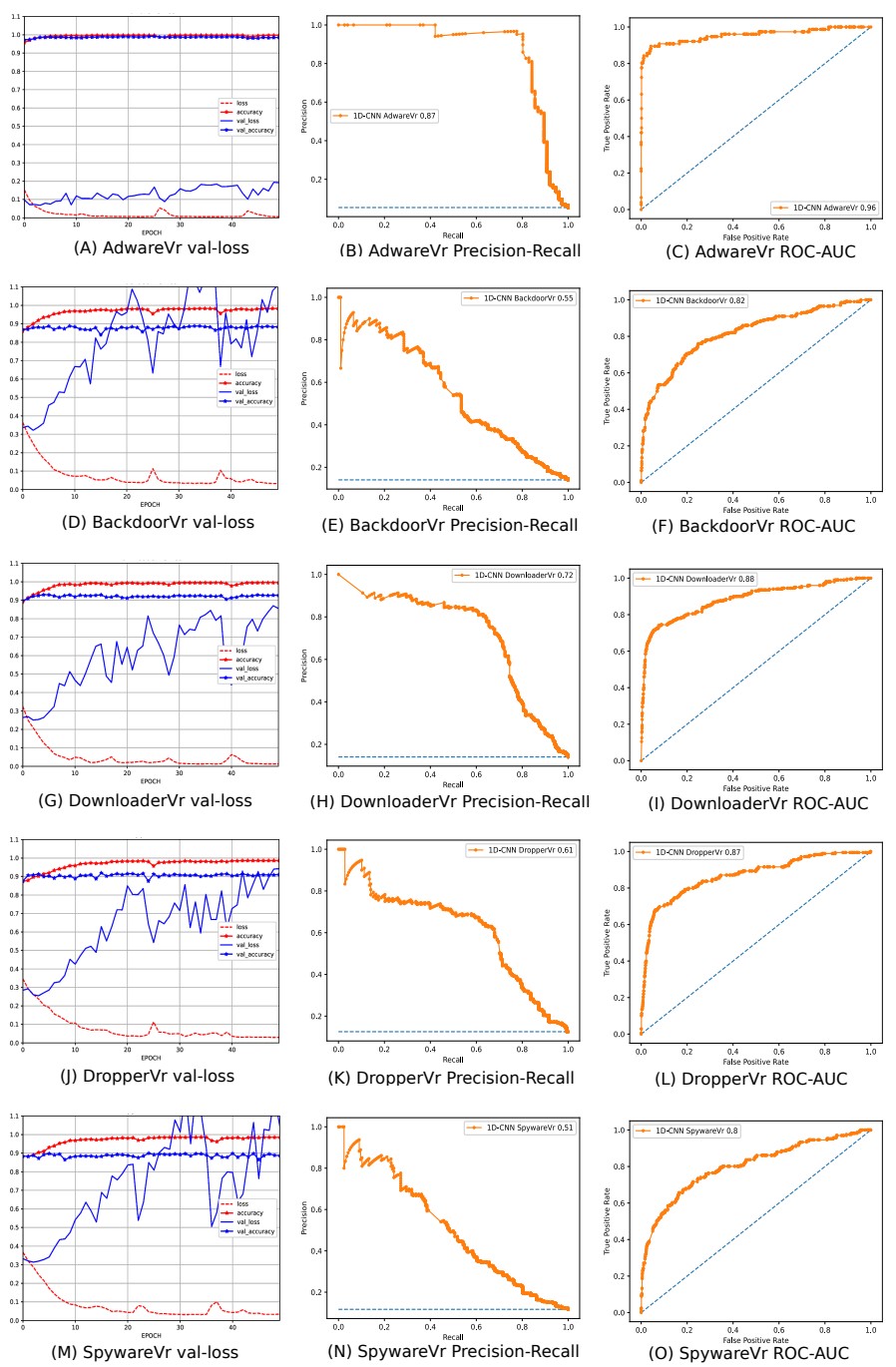

**Figure 7** **(A–O) Performance plots of 1D-CNN classifiers (Adware, Backdoor, Downloader, Dropper, and Spyware).**

Figures 7 and 8 depicts the accuracy-loss plot during training and validation, ROC plot, and Precision-Recall plot with AUC score of all the eight individual classifiers.

The classification capabilities of all these eight trained classifiers are ensembled using the proposed *ModifiedSoftVoting* algorithm as described in Algorithm-2 and used in

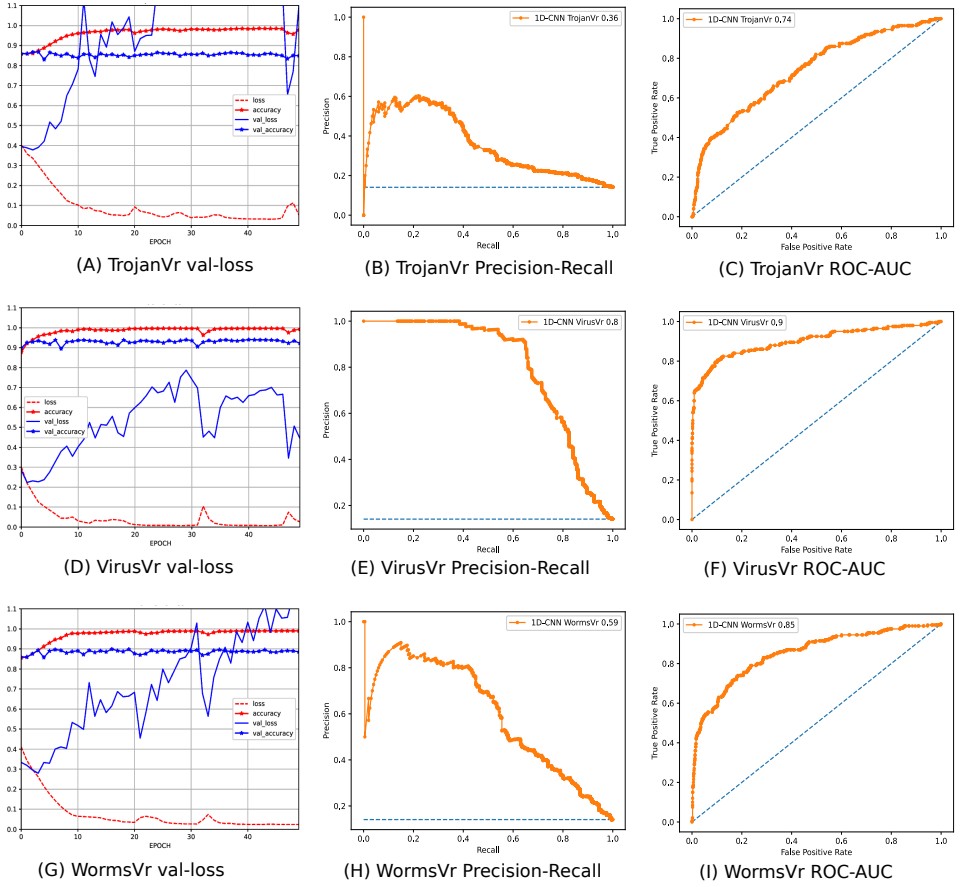

**Figure 8** (A–I) Performance plots of 1D-CNN classifiers (Trojan, Virus, and Worms).

PHASE-III of the architecture. With a stratified split rate of 20%, the multiclass *Test Set* is obtained from ProcessedDataset{ $API_{seq}$, $M_{class}$} created in PHASE-I. ModifiedSoftVoting algorithm gets the best predicted class for each record of the multiclass *Test Set*, considering the predicted results of all the 8 trained classifiers.

The statistical significance of the Ensembled ModifiedSoftVoting model is ensured by correlating its performance to a Base model. Figure 9(A) depicts a description of the design of the base model, a 1D-CNN multiclass classifier model. The confusion matrix and ROC-AUC plot of the base model are depicted in Figs. 9(B) and 9(C). In Table 4, the classification report for the ensemble ModifiedSoftVoting model is contrasted with the base model.

The performance of the base model and the ensembled ModifiedSoftVoting model is statistically correlated using a stratified sample of 20% from the MAL-API-2019 dataset in order to maintain the class distribution. The contigency table at Table 5 is constructed using the predicted labels for both models in comparison to the actual labels in the test set to conduct McNemar's test. There is a significant difference in the proportion of errors,

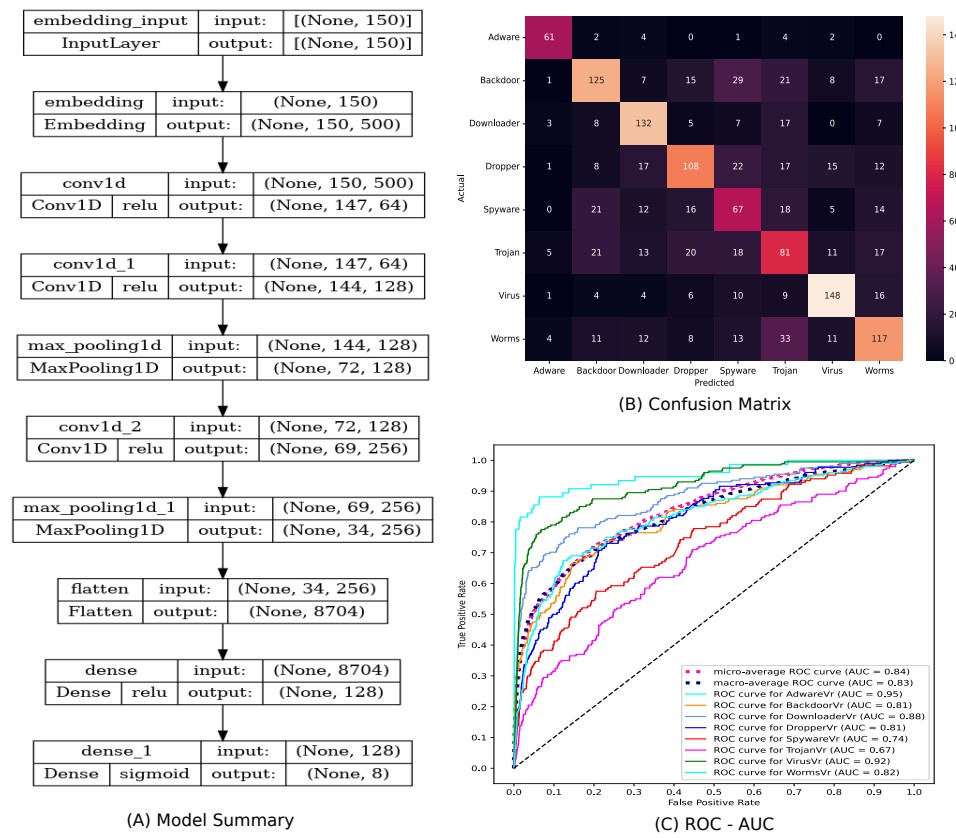

(A) Model Summary

(B) Confusion Matrix

(C) ROC - AUC

**Figure 9**  (A–C) The base model and its performance.

**Table 4**  Comparison of clssification reports.

| (A) Ensembled ModifiedSoftVoting model | | | | (B) Base Model | | | |
|---|---|---|---|---|---|---|---|
| | Precision | Recall | f1-score | | Precision | Recall | f1-score |
| Adware | 0.95 | 0.95 | 0.95 | Adware | 0.82 | 0.80 | 0.81 |
| Backdoor | 0.87 | 0.86 | 0.87 | Backdoor | 0.56 | 0.62 | 0.59 |
| Downloader | 0.99 | 0.92 | 0.95 | Downloader | 0.74 | 0.66 | 0.69 |
| Dropper | 0.97 | 0.89 | 0.93 | Dropper | 0.54 | 0.61 | 0.57 |
| Spyware | 0.77 | 0.87 | 0.82 | Spyware | 0.44 | 0.40 | 0.42 |
| Trojan | 0.85 | 0.89 | 0.87 | Trojan | 0.44 | 0.41 | 0.42 |
| Virus | 0.89 | 0.95 | 0.92 | Virus | 0.75 | 0.74 | 0.74 |
| Worms | 0.97 | 0.90 | 0.93 | Worms | 0.56 | 0.58 | 0.57 |
| | | | | | | | |
| Accuracy | | | 0.90 | Accuracy | | | 0.59 |
| Macro avg | 0.91 | 0.90 | 0.90 | Macro avg | 0.61 | 0.60 | 0.60 |
| Weighted avg | 0.90 | 0.90 | 0.90 | Weighted avg | 0.59 | 0.59 | 0.59 |

**Table 5  Contigency table for McNemar's test.**

|  | Base model correct prediction | Base model incorrect prediction |
|---|---|---|
| ModifiedSoftVoting model correct prediction | 810 | 470 |
| ModifiedSoftVoting model incorrect prediction | 29 | 113 |

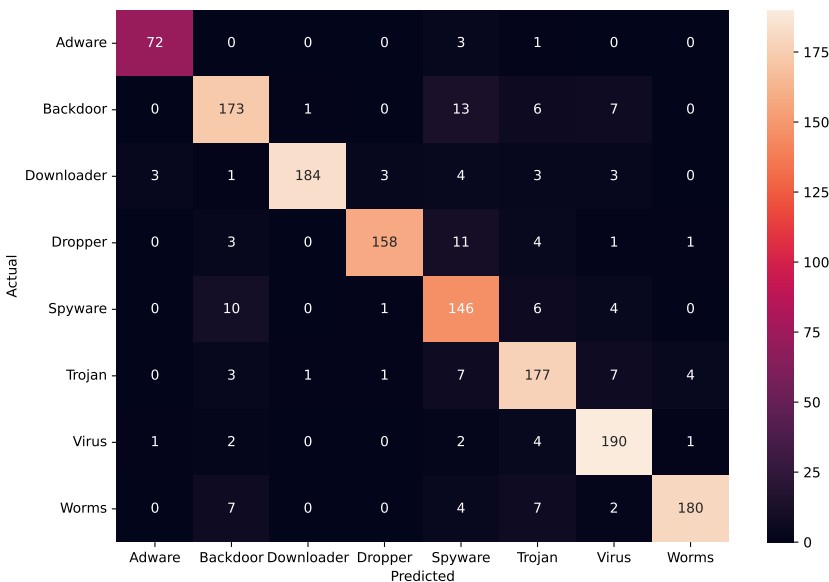

**Figure 10  Confusion matrix of ensembled ModifiedSoftVoting model.**

considering $\alpha$ as 0.05. This ensures the statistical superiority of the ensembled model over the base model.

Figure 10 depicts the confusion matrix of the classification statistics of the *Test Set* after ensembling. Figure 11A shows the ROC with AUC Score and Fig. 11B shows the Precision-Recall plot with AUC score of all the eight classes of malware present in *Mal-API-2019* using the proposed ensembled ModifiedSoftVoting model.

The performance comparison between the suggested ensembled model using the *ModifiedSoftVoting* approach and models put out by other authors in their works using the same dataset, *Mal-API-2019*, is shown in Table 6. With their suggested RTF model, *Demirkiran et al. (2022)* acquired the most recent best average $F_1$ score of 0.61. They trained a group of pre-trained transformer models known as the random transformer forest (RTF) using bootstrap sampling on the initial training set. However, our suggested ensembled model outperforms all other models with an average $F_1$ score of 0.90 and does not use bagging techniques.

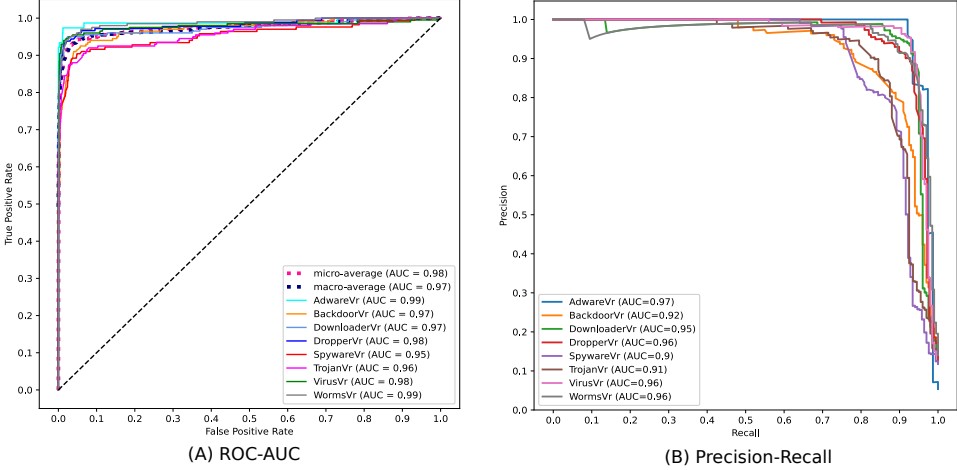

(A) ROC-AUC (B) Precision-Recall

**Figure 11  Performance plots of ensembled ModifiedSoftVoting model.**

**Table 6  Comparison of result with works of other authors.**

|  | Accuracy | Macro-avg precision | Macro-avg recall | Macro-avg F1-score |
|---|---|---|---|---|
| Single Layer LSTM (*Catak et al., 2020*) | – | 0.50 | 0.47 | 0.47 |
| Two Layer LSTM (*Catak et al., 2020*) | – | 0.40 | 0.41 | 0.39 |
| LSTM with Case2 (*Li & Zheng, 2021*) | 0.55 | 0.56 | 0.58 | 0.57 |
| GRU with Case2 (*Li & Zheng, 2021*) | 0.55 | 0.56 | 0.59 | 0.57 |
| RTF Model (*Demirkiran et al., 2022*) | 0.60 | – | – | 0.61 |
| **Proposed Ensembled Model** | **0.90** | **0.91** | **0.90** | **0.90** |

## CONCLUSION AND FUTURE DIRECTIONS

API call sequences are becoming recognized as a key characteristic for categorizing malware. To classify eight extremely unbalanced malware classes, the proposed ensembled architecture of separately trained 1D-CNN models has demonstrated good results in this work. The experimental set-up using the Mal-API-2019 benchmark dataset has demonstrated significant improvement in classification accuracy, which is now 90%. The macro averaged precision, recall, and $F_1$ score for all classes are calculated to be 91%, 90%, and 90%, respectively. The ROC plot in Fig. 11(A) displays AUC score values of 0.99, 0.97, 0.97, 0.98, 0.95, 0.96, 0.98, and 0.99, respectively, for the malware classes Adware, Backdoor, Downloader, Dropper, Spyware, Trojan, Virus, and Worms. The Precision-Recall plot in Fig. 11(B) displays AUC score values of 0.97, 0.92, 0.95, 0.96, 0.90, 0.91, 0.96, and 0.96 for each malware class, as previously mentioned. These results are encouraging and support the model's effectiveness suggested in this article. Compared to past studies' findings, which had a maximum macro average $F_1$ score of 61%, this result clearly represents a significant improvement. The impact of the data augmentation strategy on the classification outcomes for such imbalanced classes is not investigated in this work. Another approach is to examine

the effects of statistical feature engineering approaches on classification outcomes, such as PCA and duplicate subsequence removal techniques.

### Funding
The authors received no funding for this work.

### Competing Interests
The authors declare there are no competing interests.

### Author Contributions

- Binayak Panda conceived and designed the experiments, performed the experiments, analyzed the data, performed the computation work, prepared figures and/or tables, authored or reviewed drafts of the article, and approved the final draft.
- Sudhanshu Shekhar Bisoyi conceived and designed the experiments, performed the experiments, analyzed the data, performed the computation work, prepared figures and/or tables, authored or reviewed drafts of the article, and approved the final draft.
- Sidhanta Panigrahy analyzed the data, prepared figures and/or tables, authored or reviewed drafts of the article, and approved the final draft.

### Data Availability
   The Python source codes for the study are available in the Supplemental File.
   The dataset is available at GitHub and Mendeley:
   - Available at https://github.com/ocatak/malware_api_class
   - Çatak, Ferhat Özgür (2019), "Mal-API-2019", Mendeley Data, V2, doi: 10.17632/w393cchcb7.2.

### Supplemental Information
Supplemental information for this article can be found online at http://dx.doi.org/10.7717/peerj-cs.1677#supplemental-information.

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
