# Peer review of "An ensemble approach for imbalanced multiclass malware classification using 1D-CNN"

_PeerJ Computer Science, doi:10.7717/peerj-cs.1677_

## Round 0.1 · original submission · Minor Revisions

I have received reviews of your manuscript from scholars who are experts on the cited topic. They find the topic very interesting; however, several concerns must be addressed regarding discussions (statistical analysis) and methodology. These issues require a minor revision. Please refer to the reviewers’ comments listed at the end of this letter, and you will see that they are advising that you revise your manuscript. If you are prepared to undertake the work required, I would be pleased to reconsider my decision. Please submit a list of changes or a rebuttal against each point that is being raised when you submit your revised manuscript.

Thank you for considering PeerJ Computer Science for the publication of your research.

With kind regards,

Reviewer 1 ·

Basic reporting

Clear and unambiguous, professional English used throughout.
Comment: The article is mostly clear, but there are a few sentences in the introduction and discussion sections that could benefit from rephrasing for clarity.

Literature references, sufficient field background/context provided.
Comment: The literature review is comprehensive, but it would be beneficial to include recent studies from 2022 and 2023 to provide a more updated context.

Professional article structure, figures, tables. Raw data shared.
No Comment.

Self-contained with relevant results to hypotheses.
Comment: The results section adequately addresses the hypotheses.

Formal results should include clear definitions of all terms and theorems, and detailed proofs.
No Comment.

Experimental design

Original primary research within Aims and Scope of the journal.
Comment: The research aligns well with the journal's scope.

Research question well defined, relevant & meaningful. It is stated how research fills an identified knowledge gap.
Comment: The research question is well-articulated, but it would be helpful to emphasize more on how this study fills the existing knowledge gap.

Rigorous investigation performed to a high technical & ethical standard.
Comment: The methodology is sound.

Methods described with sufficient detail & information to replicate.
Comment: The methods are described in detail.

Validity of the findings

Impact and novelty not assessed. Meaningful replication encouraged where rationale & benefit to literature is clearly stated.
Comment: The rationale for replication is clear, and the study adds value to the existing literature.

All underlying data have been provided; they are robust, statistically sound, & controlled.
Comment: The statistical analysis is robust.


Conclusions are well stated, linked to original research question & limited to supporting results.
Comment: The conclusions are well-drawn and directly linked to the research question.

Additional comments

Major Comments:

Comparison with Other Works: Table 5 provides a comparison of results with other authors. It would be helpful to have a discussion section elaborating on why the proposed ensemble model outperforms other methods, especially the RTF Model (Demirkiran et al., 2022) which seems to be the closest in performance.

Methodology: The ensemble architecture of separately trained 1D-CNN models is mentioned. However, more details on the individual models, their architectures, and how they were combined would provide clarity.

Data Augmentation: The conclusion section hints at the potential impact of data augmentation strategies. Given the imbalanced nature of the dataset, it would be crucial to understand if and how data augmentation was employed in the current study.

Minor Comments:

Typographical Errors: There seems to be a typographical error in the section title "CONCULSION AND FUTURE DIRECTIONS." It should be corrected to "CONCLUSION AND FUTURE DIRECTIONS."

Reference Formatting: Ensure that all references are consistently formatted. For instance, some references provide the volume and page numbers, while others don't.

Clarification on AUC Score: The document mentions an AUC score greater than 95% in the ROC plot. It would be beneficial to provide the exact AUC value.

Reviewer 2 ·

Basic reporting

This paper present a study of a multiclass classification problem utilizing an imbalanced data set, and the characteristics used to examine the categorization of each form of malware are the API sequences from various malware classes. To combine the output of various 1D-CNN classifiers trained using the One-vs-rest principle, a 1D-CNN based ensembled architecture is presented. For training and testing purposes, the data set Mal-API-2019 published by Ozgur et al. (2020) is used.
The Word2Vec embedding technique with the Skip-gram model is used in the proposed ensembled 1D-CNN architecture to look into the semantic relationships between APIs in API sequences. A few 1D-CNN classifiers are trained on the Mal-API-2019 utilizing the One-vs-rest notion for classifying the various classes of malware. A suggested ensembling method combines each of these results to explore efficiency improvement.
The aforementioned architecture has three phases and conducts training and testing using the Mal-API-2019 data set. The dataset is vectorized in the first phase, independent 1D-CNN models are trained as One-vs-rest classifiers in the second phase, and in the final phase, an ensembled model using the ModifiedSoftVoting algorithm is created to address the multiclass malware classification problem.

Experimental design

The paper presents experimental results

Validity of the findings

The results are compared against other approaches

Additional comments

Minor remarks:
1. (Line 44). Kaspersky Security Bulletin Report 2021 is not written as a reference
2. (Line 46). The reference (kas, 2021) does not exist
3. (Line 48). The reference (Son, 2023) does no exist
4. (Line 72). The reference Ferhat Ferhat Ozgur et al. (2020) does no exist
5. The reference (line 303): Ferhat Ozgur, C., Ahmet Faruk, Y., Ogerta, E., and Javed, A. (2020). Deep learning based sequential model for malware analysis using windows exe api calls. PeerJ Comput Science 6:e285.
Must be written as, see https://peerj.com/articles/cs-285/
Catak FO, Yazı AF, Elezaj O, Ahmed J. 2020. Deep learning based Sequential model for malware analysis using Windows exe API Calls. PeerJ Computer Science 6:e285
6. (Line 97). The reference (big 2015) does no exist
7. Insert a space between words:
a) line 76. section.Comparison
b) line 77. section.The
c) line 97. kaggle(big 2015)
d) line 110. productivity.Using
e) line 166. detection(Panda
f) line 171. ones(Mikolov
g) line 221. Eq.(3,
h) line 257. layers,and
i) Table 5: LSTM(Ferhat; Case2(Li
8. Please correct the following words:
a) line 256. frequecny
b) line 244. reffered
9. Please consistently use the page numbering in the references section, in some cases you use :208-233 in others you use pages 137-149

Reviewer 3 ·

Basic reporting

The writing is clear and unambiguous. Professional English is used throughout. The literature references could be improved. The problem is motivated in Kavak et al. (Simulation for cybersecurity: state of the art and future directions). Identifying that this issue is well known as a future direction research of cybersecurity would better establish the need for the research. The source code for the paper is shared but the raw data for the paper is not shared. It is unclear why the authors hypothesized why 1D-CNN classifiers trained using the one-vs-rest principle would be effective for classifying API sequences
from various malware classes in Mal-API-2019 published by Ferhat Ferhat Ozgur. Being more specific about this would improve the paper. The terms and results in the paper are clear and are well defined.

Experimental design

The original primary research is within the aims and scope of the journal. The research is well defined and meaningful. The extent to which the stated research fills an identifiable gap could be improved. The problem is motivated in Kavak et al. (Simulation for cybersecurity: state of the art and future directions). Identifying that this issue is well known as a future direction research of cybersecurity would better establish the need for the research. The investigation is rigorous. However, the technical standard to which it is performed and the extent to which there is sufficient detail and information to replicate it would be improved by sharing the raw data used and the source code used to create the plots included in the paper

Validity of the findings

The rationale and benefit of the study could be improved by establishing the need with the aforementioned paper (Kavak et al.) which identifies it as a need / future direction. In addition, replication is difficult given that the data is not supplied and the source code related to producing the plots and tabular data. The validity of the findings would also be improved if statistical significance of the superior performance of the ensemble model compared to alternatives was established. This would demonstrate that the ensemble approach's superior performance is likely to generalize to different benchmarks as opposed to it just being noise in the evaluation data leading to ensemble approach being evaluated as more effective. The conclusions are well stated and linked to the original research question. However, limitations with the extent to which the research will generalize would improve the paper. In addition identifying internal and external validity threats to the conclusions would improve the paper.

---

## Round 0.2 · Minor Revisions

Some concerns raised by the reviewers have been partially addressed; the manuscript still needs further work regarding the statistical analysis of the proposed approach and some remarks in the references. These issues require a minor revision. If you are prepared to undertake the work required, I would be pleased to reconsider my decision. Please submit a list of changes or a rebuttal against each point that is being raised when you submit your revised manuscript.

Reviewer 1 ·

Basic reporting

All of my earlier comments have been successfully addressed by the authors.

Experimental design

All of my earlier comments have been successfully addressed by the authors.

Validity of the findings

All of my earlier comments have been successfully addressed by the authors.

Additional comments

All of my earlier comments have been successfully addressed by the authors.

I am recommending the paper to be accepted.

Reviewer 2 ·

Basic reporting

This paper present a study of a multiclass classification problem utilizing an imbalanced data set, and the characteristics used to examine the categorization of each form of malware are the API sequences from various malware classes. To combine the output of various 1D-CNN classifiers trained using the One-vs-rest principle, a 1D-CNN based ensembled architecture is presented. For training and testing purposes, the data set Mal-API-2019 is used.
The Word2Vec embedding technique with the Skip-gram model is used in the proposed ensembled 1D-CNN architecture to look into the semantic relationships between APIs in API sequences. A few 1D-CNN classifiers are trained on the data set utilizing the One-vs-rest notion for classifying the various classes of malware. A suggested ensembling method combines each of these results to explore efficiency improvement.
This architecture has three phases and conducts training and testing using data set. The dataset is vectorized in the first phase, independent 1D-CNN models are trained as One-vs-rest classifiers in the second phase, and in the final phase, an ensembled model using the ModifiedSoftVoting algorithm is created to address the multiclass malware classification problem.

Experimental design

Good design

Validity of the findings

Good results

Additional comments

Remarks:
Most of the remarks of the first review have been taken into account by the authors.
However, there are still ignored remarks in the references section such as:

9. Please consistently use the page numbering in the references section, in some cases you use: 208-233 in others you use pages 137-149

For example:
1. The reference in your paper:
Aslan, O. A. and Samet, R. (2020). A comprehensive review on malware detection approaches. IEEE Access, 8:6249–6271.
But, the IEEE Xplore recommends to cite it as:
Ö. A. Aslan and R. Samet, "A Comprehensive Review on Malware Detection Approaches," in IEEE Access, vol. 8, pp. 6249-6271, 2020, doi: 10.1109/ACCESS.2019.2963724.

2. The reference in your paper:
Kolosnjaji, B., Zarras, A., Webster, G., and Eckert, C. (2016). Deep learning for classification of malware system call sequences. In AI 2016: Advances in Artificial Intelligence, pages 137–149. Springer International Publishing.

But the Springer link recommends to cite it as:
Kolosnjaji, B., Zarras, A., Webster, G., Eckert, C. (2016). Deep Learning for Classification of Malware System Call Sequences. In: Kang, B., Bai, Q. (eds) AI 2016: Advances in Artificial Intelligence. AI 2016. Lecture Notes in Computer Science, vol 9992, pp. 137-149. Springer, Cham.
3. The reference in your paper:
Mehra, V., Jain, V., and Uppal, D. (2015). Dacomm: Detection and classification of metamorphic malware. In 2015 Fifth International Conference on Communication Systems and Network Technologies, pages 668–673.

But, the IEEE Xplore recommends to cite it as:
V. Mehra, V. Jain and D. Uppal, "DaCoMM: Detection and Classification of Metamorphic Malware," 2015 Fifth International Conference on Communication Systems and Network Technologies, Gwalior, India, 2015, pp. 668-673, doi: 10.1109/CSNT.2015.62.

4. The incomplete reference in your paper:
Vinod, P., Jain, H., Golecha, Y., Gaur, M., and Laxmi, V. (2010). Medusa: Metamorphic malware dynamic analysis using signature from api. pages 263–269.

But, the IEEE Xplore recommends to cite it as:
Vinod P. Nair, Harshit Jain, Yashwant K. Golecha, Manoj Singh Gaur, Vijay Laxmi. MEDUSA: MEtamorphic malware dynamic analysis using signature from API. SIN '10: Proceedings of the 3rd international conference on Security of information and networks September 2010, pp. 263–269, doi: https://doi.org/10.1145/1854099.1854152

5. The reference in your paper:
Zhang, Y., Huang, Q., Ma, X., Yang, Z., and Jiang, J. (2016). Using multi-features and ensemble learning method for imbalanced malware classification. In 2016 IEEE Trustcom/BigDataSE/ISPA, pages 965–973.
But, the IEEE Xplore recommends to cite it as:
Y. Zhang, Q. Huang, X. Ma, Z. Yang and J. Jiang, "Using Multi-features and Ensemble Learning Method for Imbalanced Malware Classification," 2016 IEEE Trustcom/BigDataSE/ISPA, Tianjin, China, 2016, pp. 965-973, doi: 10.1109/TrustCom.2016.0163.

6. It is highly recommended to rewrite each reference in the paper.

Reviewer 3 ·

Basic reporting

Several issues I identified have been addressed. However, the problem is still not motivated within the landscape of larger cybersecurity. Specifically in Kavak et al. (Simulation for cybersecurity: state of the art and future directions) the problem the authors are addressing is identified as a future direction research of cybersecurity would better establish the need for the research.

Experimental design

These issues have been sufficiently addressed.

Validity of the findings

Several issues I identified have been addressed. However, the problem of conducting tests for statistical significance of the superior performance of the ensemble model compared to alternatives has not been performed established. This would demonstrate that the ensemble approach's superior performance is likely to generalize to different benchmarks as opposed to it just being noise in the evaluation data leading to ensemble approach being evaluated as more effective.

---

## Round 0.3 · accepted · Accept

I am pleased to inform you that your work has now been accepted for publication in PeerJ Computer Science.

Please be advised that you are not permitted to add or remove authors or references post-acceptance, regardless of the reviewers' request(s).

Thank you for submitting your work to this journal. On behalf of the Editors of PeerJ Computer Science, we look forward to your continued contributions to the Journal.

---

## Author Rebuttal · Round 0.3

Date: 29-09-2023

Dear Editors,

At the outset, on behalf of all the authors, I thank the reviewers for their valuable comments and suggestions on the manuscript.

The manuscript's content is revised to meet the reviewers' concerns regarding the statistical analysis of the proposed approach and some remarks in the references. You'll find responses to the reviewers' comments on the following pages.

We believe that the manuscript is now suitable for publication in PeerJ Computer Science.

Dr. Sudhanshu Sekhar Bisoyi

Assistant Professor
Department of Computer Science and Information Technology
Institute of Technical Education and Research
Siksha 'O' Anusandhan (Deemed to be) University
Jagamara, Bhubaneswar-751030, Odisha, India
sudhanshu.bisoyi@gmail.com
(On behalf of all authors.)

**Reviewer 1 (Anonymous)**

*Basic reporting*
*All of my earlier comments have been successfully addressed by the authors.*

*Experimental design*
*All of my earlier comments have been successfully addressed by the authors.*

*Validity of the findings*
*All of my earlier comments have been successfully addressed by the authors.*

*Additional comments*
*All of my earlier comments have been successfully addressed by the authors.*

*I am recommending the paper to be accepted.*

**Reviewer 2 (Anonymous)**

*Basic reporting*
*This paper present a study of a multiclass classification problem utilizing an imbalanced data set, and the characteristics used to examine the categorization of each form of malware are the API sequences from various malware classes. To combine the output of various 1D-CNN classifiers trained using the One-vs-rest principle, a 1D-CNN based ensembled architecture is presented. For training and testing purposes, the data set Mal-API-2019 is used.*

*The Word2Vec embedding technique with the Skip-gram model is used in the proposed ensembled 1D-CNN architecture to look into the semantic relationships between APIs in API sequences. A few 1D-CNN classifiers are trained on the data set utilizing the One-vs-rest notion for classifying the various classes of malware. A suggested ensembling method combines each of these results to explore efficiency improvement.*

*This architecture has three phases and conducts training and testing using data set. The dataset is vectorized in the first phase, independent 1D-CNN models are trained as One-vs-rest classifiers in the second phase, and in the final phase, an ensembled model using the ModifiedSoftVoting algorithm is created to address the multiclass malware classification problem.*

*Experimental design*
*Good design*

*Validity of the findings*
*Good results*

*Additional comments*

*Remarks:*

*Most of the remarks of the first review have been taken into account by the authors.*

*However, there are still ignored remarks in the references section such as:*

*9. Please consistently use the page numbering in the references section, in some cases you use: 208-233 in others you use pages 137-149*

**Response:** The references are rewritten following the directions the esteemed reviewer gave.

**Reviewer 3 (Anonymous)**

*Basic reporting*

*Several issues I identified have been addressed. However, the problem is still not motivated within the landscape of larger cybersecurity. Specifically in Kavak et al. (Simulation for cybersecurity: state of the art and future directions) the problem the authors are addressing is identified as a future direction research of cybersecurity would better establish the need for the research.*

**Response:** We have referred to the article by Kavak et al. to establish the need for research. This is reflected in lines 145–147 of our article to ensure motivation within the larger cybersecurity landscape.

*Experimental design*

*These issues have been sufficiently addressed.*

*Validity of the findings*

*Several issues I identified have been addressed. However, the problem of conducting tests for statistical significance of the superior performance of the ensemble model compared to alternatives has not been performed established. This would demonstrate that the ensemble approach's superior performance is likely to generalize to different benchmarks as opposed to it just being noise in the evaluation data leading to ensemble approach being evaluated as more effective.*

**Response:** Lines 283-294 add further content and results to the article to highlight the ensemble model's improved performance. The ensemble model's statistical significance in comparison to a base model is confirmed by McNemar's test.